# Methods for mediation analysis with high-dimensional DNA methylation data: Possible choices and comparisons

Dylan Clark-Boucher[1], Xiang Zhou[2], Jiacong Du[2], Yongmei Liu[3], Belinda L. Needham[4], Jennifer A. Smith[4,5], Bhramar Mukherjee[2,4]*

1 Department of Biostatistics, Harvard T.H. Chan School of Public Health; Boston, Massachusetts, United States of America, 2 Department of Biostatistics, University of Michigan; Ann Arbor, Michigan, United States of America, 3 Department of Medicine, Divisions of Cardiology and Neurology, Duke University Medical Center; Durham, North Carolina, United States of America, 4 Department of Epidemiology, University of Michigan; Ann Arbor, Michigan, United States of America, 5 Survey Research Center, Institute for Social Research, University of Michigan; Ann Arbor, Michigan, United States of America

* bhramar@umich.edu

**Data Availability Statement:** Instructions for generating our simulated data can be found on our GitHub site (https://github.com/dclarkboucher/mediation_DNAm), which includes R-scripts and a

## Abstract

Epigenetic researchers often evaluate DNA methylation as a potential mediator of the effect of social/environmental exposures on a health outcome. Modern statistical methods for jointly evaluating many mediators have not been widely adopted. We compare seven methods for high-dimensional mediation analysis with continuous outcomes through both diverse simulations and analysis of DNAm data from a large multi-ethnic cohort in the United States, while providing an R package for their seamless implementation and adoption. Among the considered choices, the best-performing methods for detecting active mediators in simulations are the Bayesian sparse linear mixed model (BSLMM) and high-dimensional mediation analysis (HDMA); while the preferred methods for estimating the global mediation effect are high-dimensional linear mediation analysis (HILMA) and principal component mediation analysis (PCMA). We provide guidelines for epigenetic researchers on choosing the best method in practice and offer suggestions for future methodological development.

## Author summary

DNA methylation is an epigenetic mechanism that regulates the expression of genes, turning them "on" or "off" to meet the needs of the cell. Changes in methylation activity are associated with both health conditions and socioeconomic factors like education and access to healthcare. Recently, researchers have been interested in whether DNA methylation may act as a link between socioeconomic disadvantage and health. Standard methods to investigate whether DNA methylation is a link, or a mediator, between disadvantage and health do not work well when there are multiple mediators—in this case, DNA methylation sites—under consideration. Our study reviews 12 statistical methods for mediation analysis that can be used to analyze many methylation sites simultaneously. We compare the methods on simulated data and provide guidelines and software for their

ReadMe file that explains how to implement our simulation study. The GitHub site also includes code for implementing our DNAm data analysis, only with pseudo-data instead of the MESA data. The exact data used in the DNAm analysis can be obtained through the MESA Data Coordinating Center (https://www.mesa-nhlbi.org/) (accession: phs000209.v13.p3). Access to MESA's data requires a specific application process depending on the type of project; see https://www.mesa-nhlbi.org/ancillary.aspx and https://www.mesa-nhlbi.org/Publications.aspx for more details.

**Funding:** MESA and the MESA SHARe project are conducted and supported by the National Heart, Lung, and Blood Institute (NHLBI) in collaboration with MESA investigators. Support for MESA is provided by contracts 75N92020D00001, HHSN268201500003I, N01-HC-95159, 75N92020D00005, N01-HC-95160, 75N92020D00002, N01-HC-95161, 75N92020D00003, N01-HC-95162, 75N92020D00006, N01-HC 95163, 75N92020D00004, N01-HC-95164, 5N92020D00007, N01-HC-95165, N01-HC-95166, N01-HC-95167, N01-HC-95168, N01-HC-95169, UL1-TR-000040, UL1-TR-001079, UL1-TR-001420, UL1-TR-001881, and DK063491. The MESA Epigenomics & Transcriptomics Studies were funded by NIH grants 1R01HL101250, 1RF1AG054474, R01HL126477, R01DK101921, and R01HL135009. Co-authors of this manuscripts were partially supported by NHLBI grant R01HL141292, NSF grant DMS1712933, and NIH grants R01HG008773 and 1UG3CA267907. The funders had no role in study design, data collection and analysis, decision to publish, or preparation of the manuscript.

**Competing interests:** The authors have declared no competing interests.

implementation. We then demonstrate how the methods can be applied to real methylation data by testing whether DNA methylation sites across the genome mediate the effect of lower educational attainment on HbA1c, an important marker of type II diabetes.

## Introduction

In this study, we review and evaluate several available methods for performing mediation analysis when the mediators are high-dimensional DNA methylation (DNAm) measurements. DNAm is an epigenomic mechanism in which a methyl group binds to the DNA—a process that most often occurs at cytosine-guanine dinucleotides, called "CpG sites." One of the primary functions of DNAm is to regulate gene expression. For example, when CpG sites in the promoter regions of genes become methylated, it can discourage gene expression by inhibiting the binding of enzymes needed for transcription [1].

Advancements in modern technology have made it possible to measure DNAm on a massive scale. Indeed, microarray techniques have been used to measure more than 850,000 CpG sites at once, producing rich, detailed data that has encouraged broad research on DNAm in the etiology of disease [2]. Owing greatly to this technology and others, DNAm has been established as a risk factor in obesity [3], type II diabetes [4], schizophrenia [5], preterm birth [6], breast cancer [7], cardiovascular disease [8], and countless other conditions spanning physical and mental health. A focus of research in genetic epidemiology has been to interrogate these relationships for their predictive utility [9], biological mechanisms [10], and causality in relation to medical phenotypes [11].

However, in addition to its well-established connections to a disease or health outcome, DNAm is also associated with environmental exposures which themselves are known to affect human health. Factors such as diet [12], smoking [13], alcohol [14], air pollution [15], and socioeconomic status (SES) [16] are only a handful of the many environmental exposures that have been shown to be associated with differences in DNAm. As each of these traits have their own health risks, there have been mounting hypotheses that DNAm serves as a conduit through which assaults from the exposome are able to affect health. Effect transmission of this nature is called *mediation*, and it has become popular in epigenetic research to treat DNAm as a mediator between environmental exposures and human disease [17].

As an example of such an analysis, our previous work [18] showed associations between low SES and glycated hemoglobin (HbA1c) in the Multi-Ethnic Study of Atherosclerosis (MESA), a United States population-based longitudinal study [19]. Indicators of SES, such as education level, are strong predictors of type II diabetes [20], while HbA1c is an important risk factor of cardiovascular disease and a critical biomarker in type II diabetes diagnosis [21]. Since education level is also associated with DNAm [16,22,23], and DNAm itself with HbA1c level [24], we hypothesized that if low education results in greater HbA1c, part of that effect could be mediated by DNAm (Fig 1). The present study revisits this hypothesis for the purpose of illustration. Our sample from MESA has 963 individuals and includes DNAm measurements at 402,339 CpG sites, none of which we know for certain are related to education or HbA1c in advance.

The standard statistical tool for addressing such a hypothesis is mediation analysis. Formally, mediation is when an exposure, say $A$, affects an outcome, $Y$, in part through its effect on a single mediating variable $M$. When $M$ is a mediator of the $A$ to $Y$ association, the total effect of $A$ on $Y$ has two components: an *indirect effect*, from $A$ affecting $M$ and $M$ affecting $Y$, and a *direct effect*, from $A$ affecting $Y$ independently of $M$. In the "traditional mediation

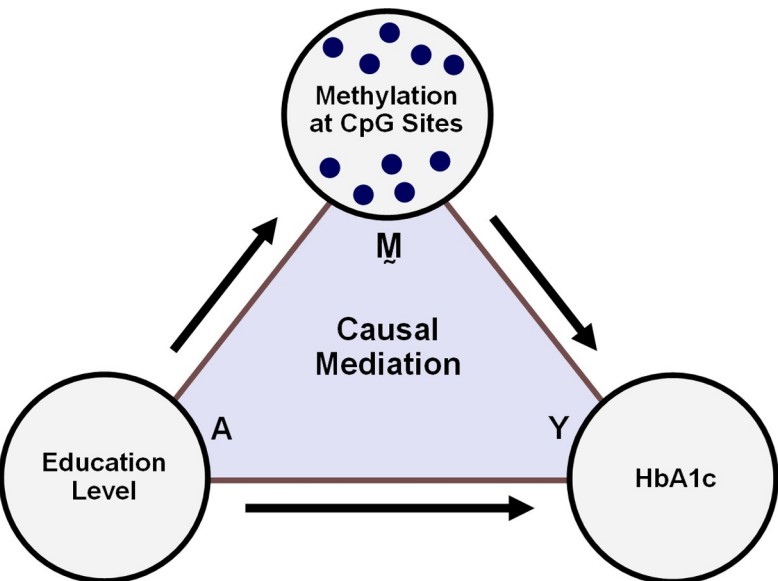

**Fig 1. Proposed causal mechanism in which the effect of low education on HbA1c is mediated by DNAm.**

analysis" approach proposed by Baron and Kenny (1986), the associations from this mechanism could be measured by fitting linear regression models: one for the effect of $A$ on $M$ (the mediator model), one for the effects of $A$ and $M$ on $Y$ (the outcome model), and sometimes a third model for the total effect of $A$ on $Y$, $M$ ignored [25–27]. The more recently developed "causal mediation analysis," based on the counterfactual approach [28,29], has established conditions under which the parameters of these models can be interpreted as causal effects [30]. The causal approach is more flexible when $Y$ or $M$ are binary and when there is $A$-$M$ interaction in the outcome model [31].

While standard examples of mediation consider only one exposure, one mediator, and one outcome [18,32], there has been growing interest in methods for mediation that can handle many potential mediators at once. Epigenetic studies have felt this need especially, as DNAm is usually measured at several hundred thousand CpG sites with little prior knowledge of their importance. Moreover, although a naïve strategy in such settings would be to evaluate the potential mediators one-at-a-time, in separate models, this approach can be problematic when the mediators are correlated conditional on the exposure variable and covariates, since the resulting estimates may be biased due confounding from the co-mediators that were excluded [18]. There could also be a loss in efficiency due to lack of exploiting the joint multivariable structure. To reduce the risk of bias and to increase precision, it is better to evaluate the mediators jointly and fit a single, multivariable outcome model that adjusts the effect of each mediator for the others, rather than fitting multiple one-at-a-time models. Though several methods for fitting such a model have been presented in the literature, they have yet to be widely adopted by practitioners and researchers for investigating substantive questions on high-dimensional mediation analysis with DNAm.

Our study aims to bridge this gap and guide researchers in epigenetics to use state of the art methods for mediation analysis with high-dimensional mediators. Despite the recent methodological developments, there are no clear-cut standards for which methods should be applied in which circumstances, making it difficult to select the best-suited method for an analysis in advance. While our prior research examined methods for large scale single-mediator hypotheses [18], there is no such work for methods that can simultaneously incorporate many

potential mediators at once. Our study first addresses this question with a simulation study, directly comparing the performance of seven different methods for mediation analysis with high-dimensional mediators across a spectrum of settings. Along with metrics related to identification of key mediators and estimation of mediation effect, we include a computation time comparison that tests the scalability of the methods to large datasets. In addition, to assess the utility of these methods on real, large-scale DNAm data, we apply the same seven methods from the simulation study, plus two additional methods, on the DNAm data provided by MESA, where we evaluate the mediating role of DNAm in the association between low education level and HbA1c. Our study is the first to address this critical gap in the applied epigenetic literature, both by providing clarity on the available methods and by assessing their strengths and weaknesses under real and simulated conditions. Although the focus of our study is applications involving DNAm, the methods explored in the text are not specific to epigenetics, and our results and guidelines should be similarly useful for researchers studying high-dimensional mediation problems in other fields.

Another key feature of our study is the presentation of a versatile, user-friendly, and well-documented R package for implementing the methods described in the text. Computer code for the methods has been made available previously, but is found in a varyingly functioning and de-centralized state across the many repositories, supplementary files, and R packages assembled by the methods' authors. Our work centralizes these resources into a single, stand-alone R package *hdmed* (https://cran.r-project.org/package=hdmed), which has the flexibility to apply multiple methods for high-dimensional mediation analysis in one place. It is our hope that by synthesizing these methods into a confined, usable package, we will catalyze the translation of our study and findings into practical, insightful research pursuits in genetic epidemiology and other fields.

## Notations and general framework

Before proceeding, it will be useful to provide an overview of the relevant mediation model and to summarize the types of methods which have become available. To begin, suppose we have a dataset of $n$ individuals: an exposure $A_i$, a continuous outcome $Y_i$, and continuous mediators $M_i$ measured for the $i^{\text{th}}$ person, $i$ varying from 1 to $n$. We write $M_i$ in bold to indicate its status as a vector—in this case, a set of $p$ mediators $M_i^{(j)}$, $j$ varying from 1 to $p$. Let $C_i$ be a vector of $q$ covariates. When $p$ is greater than 1 (and possibly greater than $n$), we can use the regression models

$$E[Y_i|A_i, M_i, C_i] = \beta_a A_i + \beta_m^T M_i + \beta_c^T C_i \tag{1}$$

and

$$E[M_i|A_i, C_i] = \alpha_a A_i + \alpha_c C_i \tag{2}$$

to estimate the mediating role of $M_i$ in the causal pathway between the exposure and outcome [33]. Model (1) is the outcome model and model (2) is the mediator model. In model (1), $\beta_m$ is a $p$-vector in which the $j^{\text{th}}$ component, $(\beta_m)_j$, is the linear association of $j^{\text{th}}$ mediator with $Y_i$ adjusting for the other variables; while $\beta_a$ is the association between $A_i$ and $Y_i$ adjusting for mediators and covariates. In model (2), $\alpha_a$ is a $p$-vector of the associations between the exposure and each mediator, $(\alpha_a)_j$; and $\alpha_c$ is a matrix of the mediator-covariate associations. Also note that in model (1), we have assumed there is no interaction between $A_i$ and $M_i$, which is beyond the scope of our present study.

The parameters of these models underly the causal effects of interest. Under certain assumptions (Section 1 in S1 Text) [28,33], the direct effect of $A_i$ on $Y_i$ is $\beta_a$, the global indirect

effect (or global mediation effect) of $A_i$ on $Y_i$ through $\boldsymbol{M}_i$ is $\boldsymbol{\alpha_a}^T\boldsymbol{\beta_m}$, and the total effect of $A_i$ on $Y_i$ is $\beta_a + \boldsymbol{\alpha_a}^T\boldsymbol{\beta_m}$. Another quantity of interest is the proportion mediated, defined as the ratio of the global indirect effect to the total effect, which measures the degree to which the $A_i$ to $Y_i$ pathway is mediated by $\boldsymbol{M}_i$. Lastly, we may also seek to measure the product terms $(\boldsymbol{\alpha_a})_j(\boldsymbol{\beta_m})_j$, which we will call the *mediation contributions*. The mediation contribution of the $j$th mediator reflects the mathematical contribution of that mediator to the global mediation effect, since the sum of $(\boldsymbol{\alpha_a})_j(\boldsymbol{\beta_m})_j$ over all $j$ equals $\boldsymbol{\alpha_a}^T\boldsymbol{\beta_m}$. These parameters are intuitive to estimate, but difficult to interpret. Though it is tempting to refer to $(\boldsymbol{\alpha_a})_j(\boldsymbol{\beta_m})_j$ as a causal effect corresponding to the $j$th mediator, we emphasize that this parameter cannot generally be interpreted as the natural indirect effect through that mediator specifically. Identifying the indirect effects of specific mediators, in settings with multiple mediators, requires strong assumptions about whether the group of mediators are sequentially ignorable—conditions that would be violated, for example, in situations where a subset of mediators have causal effects on some of the others. (The exact assumptions are not described here as they would require a discursion into counterfactual inference. See [34]). Despite the limited interpretability of the mediation contributions, we will refer to a mediator as *inactive* if its mediation contribution is zero, and *active* otherwise. This has the caveat that if a mediation contribution is zero, that mediator could still be involved in the causal path from $A$ to $Y$, since complex causal relationships among the set of mediators might exist.

If the potential mediators are uncorrelated, conditional on the exposure and covariates, or if $p$ is reasonably small relative to $n$, then it is trivial to fit the above models using linear regression. However, if the mediators are correlated and $p$ is large, the estimates from model (1) may have extremely high variance; and if $p$ is so large as to exceed $n$, the linear regression model cannot even be fitted. These concerns are relevant to us because DNAm measurements tend to be correlated, while the number of sites that we have measurements on exceeds the number of samples. Addressing these issues has been a recent focus of the mediation literature, with authors using penalized regression [35–40], dimension reduction [41–43], Bayesian inference [44,45], and latent variables [46] to make the outcome model statistically tractable.

We provide a graphical depiction of 12 available methods in Fig 2. Eight of them are assessed in the simulation study, ten of them are used in the DNAm analysis, and all of them are described in the Methods section. To help elucidate the differences between methods, we partition them into three distinct groups based on their approaches and objectives. In the first group, we consider methods that explicitly fit the outcome and mediator models as we have defined them so that one can estimate $\boldsymbol{\alpha_a}^T\boldsymbol{\beta_m}$, the global indirect effect, simply by summing the estimates of the mediation contributions. The methods for doing so are *high-dimensional mediation analysis* (HIMA) by Zhang et al. 2016 [35], *high-dimensional mediation analysis* (HDMA) by Gao et al. 2019 [36], *mediation analysis via fixed effect model* (MedFix) by Zhang 2019 [37], *pathway least absolute shrinkage operator* (pathway LASSO) by Zhao and Luo 2022 [38], the *Bayesian sparse linear mixed model* (BSLMM) by Song et al. 2020 [44], and the *Gaussian mixture model* (GMM) by Song et al. 2021 [45]. In contrast, the second group of methods considers those that can estimate $\boldsymbol{\alpha_a}^T\boldsymbol{\beta_m}$ "directly"—without needing to fit the mediation models we began with in their original form. These methods have the drawback of being unable to estimate the mediation contributions of specific active mediators. They include *principal component mediation analysis* (PCMA) by Huang and Pan 2016 [41], *sparse principal component mediation analysis* (SPCMA) by Zhao et al. 2020 [42], *high-dimensional linear mediation analysis* (HILMA) by Zhou et al. 2021 [39], and a method we will call *partial penalized high-dimensional mediation analysis* (PMED), proposed by Guo et al. 2022 [40]. Lastly, the third group of methods are those that make no attempt to estimate the mediation effects as originally proposed, but that instead reconceptualize the mediation framework with newly-defined

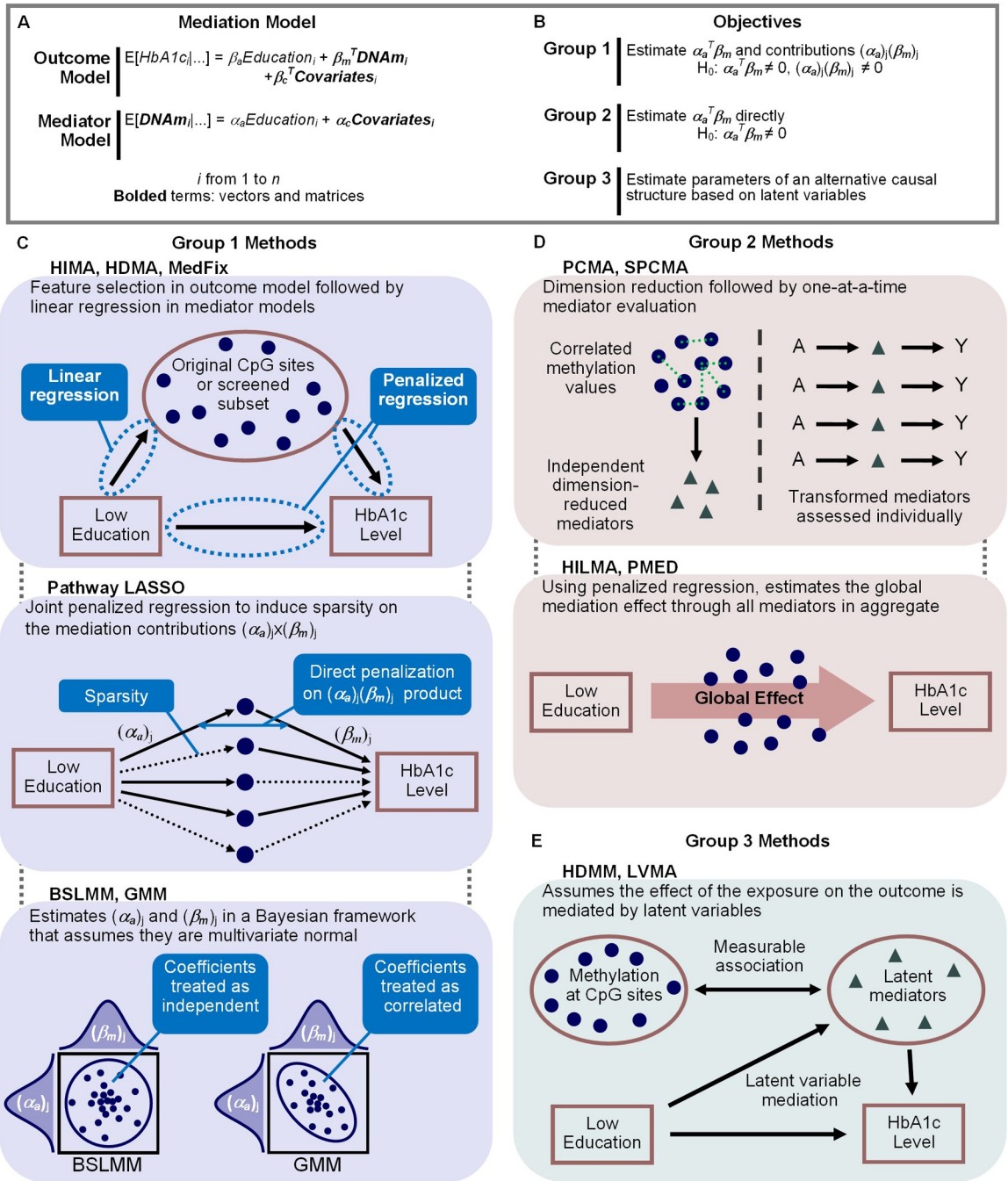

**Fig 2. Methods for mediation analysis with high-dimensional DNAm data.** (A) Statistical methods for high-dimensional mediation analysis require a multivariable outcome model and multivariate mediator model. (B) Group 1 methods estimate the global mediation effect ($\alpha_a^T\beta_m$) by fitting the outcome model and estimating the mediator-specific contributions; Group 2 methods estimate $\alpha_a^T\beta_m$ directly without fitting the original model; and Group 3 methods estimate the parameters of an alternative causal structure based on latent variables. (C) In Group 1, the methods HIMA, HDMA, and MedFix apply penalized regression to the outcome model and then linear regression to the mediator model; the method Pathway LASSO fits the outcome and mediator model simultaneously with a jointly penalized likelihood; and the Bayesian methods BSLMM and GMM use multivariate normal mixture models. (D) In Group 2, the methods PCMA and SPCMA use principal component analysis to replace the observed, correlated mediators with independent mediators that can be assessed one-at-a-time. The method HILMA uses a multi-step penalized regression procedure that estimates $\alpha_a^T\beta_m$ but not the mediation contributions. (C) In Group 3, the methods HDMM and LVMA construct latent mediators which replace the original mediators in the mediation model, and thus, they do not yield estimates of $\alpha_a^T\beta_m$.

parameters based on latent variables. The methods include *high-dimensional multivariate mediation analysis* (HDMM) by Chén et al. 2018 [43] and *latent variable mediation analysis* (LVMA) by Derkach et al. 2021 [46]. Within this comparative structure, we evaluate methods from all three groups, identifying their strengths and weaknesses across a wide range of simulation settings and analysis of DNAm data from MESA.

## Materials and methods

### Overview of methods

Let $A$ be an exposure, $Y$ be a continuous outcome, and $M$ be a set of $p$ continuous variables that potentially mediate the causal path from $A$ to $Y$. Then with $A_i$, $Y_i$, $M_i$, and covariates $C_i$ measured for $n$ subjects, $i$ from 1 to $n$, we can evaluate the mediating role of $M$ with models (1) and (2) as presented in the Introduction section. We provide an overview of 12 methods for mediation analysis that can accommodate this multivariate framework below. A tabular summary is given in the supplement (S1 Table).

### Group 1 methods: Penalized regression to estimate mediator-specific contributions

**HIMA.**   High-dimensional mediation analysis (HIMA) by Zhang et al. (2016) is a penalized regression approach in which the outcome model is fit with a minimax concave penalty [47], performing feature selection on the potential mediators [35]. The mediator models are then fit among the remaining mediators using ordinary linear regression. Finally, the "significance" of the mediation contributions is tested by taking the maximum of the $(\beta_m)_j$ and $(\alpha_a)_j$ p-values, where the p-values for $(\beta_m)_j$ are obtained by refitting the reduced outcome model with ordinary least squares (an approach which is likely to cause the p-values to be overconfident [48]). The authors also recommend an initial screening step to reduce the number of mediators at the start, as the outcome model will still be unstable if $p$ is extremely large compared to $n$. A new version of HIMA, called HIMA2, was published recently [49]. HIMA2 is similar to HDMA, but suggests a p-value correction procedure that maintains the false discovery rate for detecting active mediators. HIMA2 is excluded from our comparison due to its similarity to HDMA.

**HDMA.**   High-dimensional mediation analysis (HDMA) by Gao et al. (2019), is the same as HIMA except for its penalty function, replacing the minimax concave penalty with the recently-proposed de-sparsified LASSO [36,50]. The advantage of this penalty is that the resulting estimates of $\beta_m$ are asymptotically normally distributed, so one can test their statistical significance without a subsequent application of ordinary least squares.

**MedFix.**   Mediation analysis via fixed effect model (MedFix) is another extension of HIMA, proposed by Zhang (2021) [37]. MedFix was originally proposed for a setting where there are multiple exposures in addition to multiple mediators, which it handles by applying adaptive LASSO [51] to both the outcome model and the mediator models. If there is only one exposure, linear regression can replace adaptive LASSO in the mediator models, and applying MedFix is analogous to applying HDMA except with adaptive LASSO instead of debiased LASSO.

**Pathway LASSO.**   Pathway LASSO is a penalized regression approach by Zhao and Luo (2022) [38]. Whereas HIMA, HDMA, and MedFix handle the outcome and mediator models separately, this method fits the models simultaneously with a jointly penalized likelihood that directly applies shrinkage to the mediator-outcome associations, exposure-mediator associations, and their products (the mediation contributions).

**BSLMM.** The Bayesian sparse linear mixed model (BSLMM) by Song et al. (2020) assumes $\boldsymbol{\alpha}_a$ and $\boldsymbol{\beta}_m$ are random vectors that independently follow mixtures of normal distributions [44]. Most of the effects are assumed to be small, resulting from a normal distribution with low variance, while a minority are assumed to be larger and follow a normal distribution with higher variance. Active mediators are discriminated from inactive by their posterior inclusion probability of belonging to the higher-variance distribution.

**GMM.** The Gaussian mixed model (GMM) by Song et al. (2021) is an extension of BSLMM in which the $(\boldsymbol{\alpha}_a)_j$, $(\boldsymbol{\beta}_m)_j$ pairs are treated as correlated, following a mixture of multivariate normal distributions instead of two independent normal distributions [45]. Thus, GMM may be more useful than BSLMM if the true size of each $(\boldsymbol{\beta}_m)_j$ is related to the size of the corresponding $(\boldsymbol{\alpha}_a)_j$, and vice-versa.

## Group 2 methods: Dimension reduction and direct estimation of global indirect effect

**PCMA.** Principal component mediation analysis (PCMA) by Huan and Pan (2016) is a mediation analysis method based on principal component analysis (PCA) [41]. The authors perform PCA on the residual matrix of the mediator models, then use the resulting loading matrix to transform $\boldsymbol{M}$ into a new set of mediators which are uncorrelated conditional on $A$ and $\boldsymbol{C}$. The transformed mediators then replace the original mediators in the analysis and are evaluated in a one-at-a-time fashion. In spite of the transformation, the global indirect effect $\boldsymbol{\alpha}_a^T\boldsymbol{\beta}_m$ can still be estimated with its original interpretation as the global mediation effect through $\boldsymbol{M}$. The authors set the number of transformed mediators to be $p$, though this is only possible if $p$ is less than $n$.

**SPCMA.** Zhao et al (2019) proposed sparse principal component analysis (SPCMA) to improve the interpretability of the results from PCMA [42]. In PCMA, the transformed mediators are difficult to interpret because they are sums of all $p$ original mediators; whereas in SPCMA, the loading matrix is sparsified so that the transformed mediators are only sums of only a subset of the original mediators. Thus, if a specific transformed mediator has a large effect, it can potentially be traced back to the original mediators which were used to construct it. Though the added sparsity induces bias, it can be helpful for identifying groups of mediators which may be active.

**HILMA.** High-dimensional linear mediation analysis (HILMA) by Zhou (2020) estimates $\boldsymbol{\alpha}_a^T\boldsymbol{\beta}_m$ with a complex, de-biased penalized regression procedure that is beyond the scope of this article [39]. The proposed estimator has asymptotic properties for testing whether $\boldsymbol{\alpha}_a^T\boldsymbol{\beta}_m$ is zero and can also be applied when there are multiple (but not more than $n$) exposures.

**PMED.** Partial penalized high-dimensional mediation analysis (PMED) is a two-step estimation and inference procedure for the global mediation effect, proposed by Guo et al. (2022) [40]. In the first step, the outcome model is fitted with the mediators penalized by the smoothly-clipped absolute deviation (SCAD) penalty. In the second step, the estimated direct effect from the outcome model is subtracted from an estimated total effect, which is obtained by fitting an unpenalized outcome model with the mediators omitted. The method reports the global mediation effect and a set of potentially active mediators selected in step one, but does not provide estimates of the mediator-specific mediation contributions. PMED can also be applied when there are multiple, but fewer than $n$, exposure variables.

## Group 3 methods: Latent variable representation to summarize mediators

**HDMM.** High-dimensional multivariate mediation (HDMM) by Chén et al. (2018) uses dimension reduction similar to PCMA, but chooses the loading vectors with a likelihood-

based approach instead of PCA [43]. The loading vectors are referred to as "directions of mediation," each vector specifying a linear combination of mediators which contribute to the likelihood of the mediation models. This implicitly assumes there are latent, unmeasured mediators that can be represented as linear combinations of the observed mediators. The results of HDMM are difficult to interpret, but it can still be useful for identifying whether there is any mediation through $M$ at all and for identifying large subsets of mediators that contribute to that mediation. A limitation of HDMM is that it cannot directly be applied when $p$ exceeds $n$.

**LVMA.** Latent variable mediation analysis (LVMA) by Derkach et al. (2019) assumes are a small number of latent, unmeasured mediators $F$ which transmit the effect of $A$ to $Y$ and which also cause changes in $M$ [46]. Thus, LVMA assumes explicitly what HDMM assumes implicitly, and the results of the methods have a similar interpretation. Another feature of LVMA is that the $F \rightarrow M$ associations are sparsified, which is useful for detecting relevant mediators in $M$. Indeed, an observed mediator would be considered active if it is associated with a latent mediator that itself is associated with $A$ and $Y$.

## Simulation study

### Simulation settings

**Primary simulation settings.** We evaluated the above methods with a simulation study. To contrast them under diverse conditions, we considered three different settings of mediation: (1) a baseline setting in which the mediation signals are sparse and the error terms of the (potential) mediators are moderately correlated, (2) a high-correlation setting with sparse signals, and (3) a moderate correlation setting in which the signals are non-sparse. We also varied the signal strength of the mediation by modifying three parameters: the proportion of variance explained by $A$ in mediators affected by $A$ ($\mathrm{PVE_A}$); the proportion of variance in $Y$ explained by the direct effect ($\mathrm{PVE_{DE}}$); and the proportion of variance in $Y$ explained by the global indirect effect ($\mathrm{PVE_{IE}}$). For a baseline case, we let $\mathrm{PVE_A}$ be 0.20, $\mathrm{PVE_{DE}}$ be 0.1, and $\mathrm{PVE_{IE}}$ be 0.10. Then, for three additional cases, we sequentially decreased one of these parameters by half, weakening the signal, and set the other two parameters to their values from the baseline. Each of the four signal strengths was evaluated in each of settings (1) to (3) with a sample size of 1,000 and 2,500, with the number of mediators fixed at 2,000. This amounted to 24 simulation settings in total. A complete list of the primary simulation settings is provided in the supplement (S2 Table), as are the numerical results underlying the figures along with code for generating the simulated data (S2 and S3 Files). None of the settings adjusted for confounding variables.

**Additional simulation settings.** To broaden the variety of simulation conditions, we consider two additional sets of simulations that involve specifc changes to the data-generating mechanism. In the first additional scenario, we consider cases in which the coefficients of the outcome and mediator models are not mixed in sign, but strictly non-negative (as explained below, the coefficients in the primary simulation settings had both positive and negative signs). The non-negative effect simulations are analogous to simulation Setting (1) above, but with the coeffcents of the model converted to their absolute value. They include each of the four signal strength settings explored previously.

Finally, the second additional scenario considers data-generating mechanisms in which there is an unmeasured confounding variable, $U$, that directly influences the exposure, the outcome, and a subset of the mediators. For these simulations, we begin with Setting (1) (as described above) with the first set of signal strength parameters ($\mathrm{PVE_A} = 0.2$, $\mathrm{PVE_{DE}} = 0.1$, $\mathrm{PVE_{IE}} = 0.1$), then perturb the data-generating mechanism by adding confounding effects of $U$ to the generation of $A$, $M$, and $Y$. We explore different levels of confounding by setting the

sensitivity analysis parameter, namely the variance of $U$, to be 1, 2, or 3, while holding the effects of $U$ on the other variables constant. In both additional simulation scenarios we set $n$ to be 2,500. A list of the additional scenarios is provided in the supplement (S3 Table). Results for both scenarios are reported in the supplement as well (S1 and S2 Figs).

## Simulated dataset creation

**Primary simulation settings.** To obtain sparse mediation effects for Settings (1) and (2), we let 1,920 of the 2,000 coefficients $(\boldsymbol{\alpha_a})_j$ and $(\boldsymbol{\beta_m})_j$ be zero and the remaining 80 be standard normal. Twenty of the nonzero $(\boldsymbol{\alpha_a})_j$ and $(\boldsymbol{\beta_m})_j$ were chosen to overlap and have the product $(\boldsymbol{\alpha_a})_j(\boldsymbol{\beta_m})_j$ not equal zero. To obtain non-sparse signals for Setting (3), we sampled the previously zero coefficients from a normal distribution with mean zero and standard deviation 0.2. Once these parameters were fixed, we obtained a single simulated dataset by sampling $A_i$ from a standard normal distribution, then produce $\boldsymbol{M_i}$ from model (2) assuming there are no covariates. We add noise to $\boldsymbol{M_i}$ by sampling residuals from a multivariate normal distribution with mean $\boldsymbol{0_p}$ and variance $\boldsymbol{S}$, where $\boldsymbol{S}$ is derived by shuffling, then tuning the variance-covariance of the observed methylation data (Section 2 in S1 Text). In Settings (1) and (3), we tune $\boldsymbol{S}$ so that the error correlations between mediators range from -0.37 to 0.49, and in Setting (2), so that they range from -0.68 to 0.89. We fix PVE$_A$ by scaling $\boldsymbol{S}$ appropriately based on $\boldsymbol{\alpha_a}$. Finally, we define $Y_i$ based on model (1) assuming the residuals are Normal(0, $s^2$), choosing $\beta_a$ and $s^2$ to yield the desired PVE$_{DE}$ and PVE$_{IE}$.

**Additional simulation settings.** For the simulations with non-negative effects, the $(\boldsymbol{a_a})_j$ and $(\boldsymbol{\beta_m})_j$ coefficients from Setting (1) are converted to their absolute value. Since this also changes the global indirect effect, $\boldsymbol{\alpha_a}^T \boldsymbol{\beta_m}$, we update the direct effect, $\beta_a$, to equal $\boldsymbol{\alpha_a}^T \boldsymbol{\beta_m} \sqrt{PVE_{DE}/PVE_{IE}}$, so that the ratio of the variance of $Y$ explained by the direct effect to the variance of $Y$ explained by the indirect effect is the same as previously. No other parameters used to generate the data are modified; for example, the residual variance of the outcome model ($s^2$) is the same as before.

For the unmeasured confounding simulations, the modified data-generating mechanism is described in models (3), (4), and (5), which are shown below:

$$A_i = \gamma U_i + \delta_i \tag{3}$$

$$\boldsymbol{M}_i = \boldsymbol{\alpha}_a A_i + \boldsymbol{\alpha}_u U_i + \boldsymbol{\varepsilon}_i \tag{4}$$

$$Y_i = \beta_a A_i + \beta_u U_i + \boldsymbol{\beta}_m^T \boldsymbol{M}_i + \zeta_i \tag{5}$$

Here, $\delta_i$ and $\zeta_i$ are independent normal random variables with mean zero, and their variances are chosen to be equal to their values from the baseline setting (1 and $s^2$, respectively). In model (4), $\boldsymbol{\varepsilon}_i$ is a multivariate normal random vector, independent of $\delta_i$ and $\zeta_i$, with variance-covariance matrix set to be $\boldsymbol{S}$ from the baseline setting. The confounder-exposure effect $\gamma$ is set to be 1/3, and the confounder-outcome effect $\beta_u$ is set to $\beta_a/2$. For the vector of confounder-mediator effects, $\boldsymbol{\alpha_u}$, we set the $j$th entry to be $(\boldsymbol{\alpha_a})_j/2$ if $(\boldsymbol{\alpha_a})_j$ is not zero, and set it to be 1/2 if $(\boldsymbol{\alpha_a})_j$ is zero but $(\boldsymbol{\beta_m})_j$ is not zero. (That is, only the mediators that are affected by $A$, affect $Y$, or both, are directly affected by $U$.) The choice of these fractions (e.g., $\beta_a/2$) is somewhat arbitrary, but it ensures the confounding effects are on a similar scale to the coefficients of interest, only slightly weaker. The remaining parameters are held at their values from Setting (1). The confounding variable $U$ is sampled from a normal distribution with mean zero and variance $\tau$, the sensitivity analysis parameter, which is set to be 1, 2, or 3. This varies the intensity of the confounding.

## Simulation analysis

We performed mediation analysis on 100 simulated datasets in each setting. We omitted the methods SPCMA, GMM, and LVMA because were too computationally costly, and omitted HDMM because its estimand is not comparable to the others. We also included a one-at-a-time approach in which the mediators are assessed one-at-a-time using traditional mediation analysis and the joint significance test [52]. When running HIMA, HDMA, MedFix, and pathway LASSO, we pre-screened the mediators to only include the $n/\log(n)$ mediators most-associated with $Y$ adjusting for $A$, which is recommended by the HIMA and HDMA authors [35,36]. For comparison metrics, we used the true positive rate for detecting active mediators, $\text{TPR} = \frac{\text{number of true mediators identified}}{\text{number of true mediators}}$; the mean squared error in estimating the mediation contributions of inactive mediators, $\text{MSE}_{\text{Inactive}} = \text{mean}_{j:\text{Inactive}}((\widehat{\boldsymbol{\alpha}_a})_j(\widehat{\boldsymbol{\beta}_m})_j - (\boldsymbol{\alpha}_a)_j(\boldsymbol{\beta}_m)_j)^2$; the mean squared error in estimating the mediation contributions of active mediators, $\text{MSE}_{\text{Active}} = \text{mean}_{j:\text{Active}}((\widehat{\boldsymbol{\alpha}_a})_j(\widehat{\boldsymbol{\beta}_m})_j - (\boldsymbol{\alpha}_a)_j(\boldsymbol{\beta}_m)_j)^2$; and the percent relative bias in estimating the global indirect effect, $\frac{|\widehat{\boldsymbol{\alpha}_a^T\boldsymbol{\beta}_m} - \boldsymbol{\alpha}_a^T\boldsymbol{\beta}_m|}{\boldsymbol{\alpha}_a^T\boldsymbol{\beta}_m} \times 100$. In the non-sparse setting, "active" mediators were considered those with both effects sampled from the high-variance distribution. We provide additional details on how the methods were applied in the supplement (Section 3 in S1 Text).

## Data application with MESA

### Study design

Data were provided by the Multi-Ethnic Study of Atherosclerosis (MESA), a United States population-based longitudinal study on the progression of subclinical cardiovascular disease [19]. Briefly, MESA recruitment ran from July 2000 to August 2002 and comprised 6,814 participants ages 45 to 84. From 2010 to 2012, a subsample of 1,264 random patients had their DNAm measured at 484,882 CpG sites. Standard quality control filters reduced the number of CpGs considered to 402,339 [53]. To demonstrate an application of high-dimensional mediation analysis methods, we evaluated whether DNAm mediates the association between SES and HbA1c in MESA. For the exposure, we used a binary variable that indicates low educational attainment (less than a 4-year college degree). For the outcome, we used HbA1c, a continuous variable that reflects average three-month blood glucose level. We limit our analysis to the 963 participants who (1) had methylation data, (2) had no missing data for the required variables, (3) consented to genetic and phenotypic use through the database of Genotypes and Phenotypes (dbGaP) (phs000209.v13.p3), and (4) were not on diabetes medication, which can cause changes in HbA1c. See supplement for more details (Section 4 in S1 Text). DNAm was measured using M-values, defined as the log-2 ratio of the methylated to unmethylated probe intensities [54].

### Statistical analysis

We performed mediation analysis with the methods HIMA, HDMA, HILMA, MedFix, pathway LASSO, PMED, BSLMM, PCMA, SPCMA, and HDMM, based on the models

$$E[\text{HbA1c}_i|\text{Education}_i, \mathbf{DNAm}_i, \mathbf{Covariates}_i]$$
$$= \beta_a\text{Education}_i + \boldsymbol{\beta}_m^T\mathbf{DNAm}_i + \boldsymbol{\beta}_c^T\mathbf{Covariates}_i \tag{6}$$

and

$$E[\mathbf{DNAm}_i|\text{Education}_i, \mathbf{Covariates}_i] = \boldsymbol{\alpha}_a\text{Education}_i + \boldsymbol{\alpha}_c\mathbf{Covariates}_i, \tag{7}$$

with the same parameters as models (1) and (2). The covariates included age, sex, race, methylation chip, methylation position, and the estimated proportions of residual non-monocytes (i.e., neutrophils, B cells, T cells, and natural killer cells). Since it is not statistically feasible to include 402,339 mediators at once, we used model (7) to select the 2,000 CpG sites most strongly associated with education based on the $(\boldsymbol{\alpha_a})_j$ p-value from a linear mixed-model in which methylation chip and position were treated as random effects. These 2,000 formed the baseline set of CpGs for our analysis. Although it is reasonable for some of the methods to include all 2,000 CpG sites directly in the multivariable model, HIMA and HDMA require sure independence screening [55] to reduce the number of mediators in advance to $n/\log(n)$, where $n$ is the sample size. For the sake of consistency across the penalized regression methods, we also do this extra screening with MedFix and pathway LASSO, including only the 141 (963/log(963)) CpG sites most associated with low education (a direct extension of the initial screening). We also use this twice-screened subset for HDMM, which requires that $p$ is less than $n$. For the sake of comparison with multivariate methods, we include a one-at-a-time mediation method based on linear mixed models and the joint significance test. For the methods PCMA, SPCMA, BSLMM, PMED, and Pathway LASSO, which produce estimates of the direct effect, the total effect is estimated by summing the direct effect and global indirect effect. For the methods HIMA, HDMA, and MedFix, which do not estimate the direct effect, we estimate the total effect by fitting model (5) with the mediators excluded, then subtract the estimated global indirect effect from this value to estimate the direct effect. Since none of the high-dimensional methods can handle random effects as covariates, we regress methylation chip and position out of $Y$ and $\boldsymbol{M}$ in advance with a linear mixed model, while fixed-effect covariates are either regressed out as well (in PCMA, SPCMA, HILMA, HDMM, and pathway LASSO) or included directly in the method (in HIMA, HDMA, MedFix, PMED, and BSLMM). Continuous variables (including HbA1c and the mediators) were standardized for all methods. The methods LVMA and GMM were too computationally costly to implement. All analysis was conducted using R version 4.2.1.

## Results

### Simulation results

We begin by comparing the performance of the methods using simulations. On simulated data with 2,000 potential mediators, we consider (1) a baseline setting, where the error terms of the mediators are moderately correlated and the signals of the mediators are sparse; (2) a high-correlation setting, where the error correlations between mediators are enhanced compared to (1); and (3) a non-sparse setting, where every mediator has at least some mediation signal but some of the signals are systematically larger. In Settings (1) and (2), 60 random mediators have $(\boldsymbol{\alpha_a})_j$ only sampled from a Normal(0,1), 60 have $(\boldsymbol{\beta_m})_j$ only sampled from a Normal(0,1), and 20 have both, with the remaining entries of $\boldsymbol{\alpha_a}$ and $\boldsymbol{\beta_m}$ fixed at zero. In Setting (3), we use a similar scheme but sample the previously zero $(\boldsymbol{\alpha_a})_j$ and $(\boldsymbol{\beta_m})_j$ from a Normal(0,0.2$^2$). Our simulations also vary the strength of the signals within each of these settings by changing the proportion of variance that is explained by the associations. We do so by changing $\text{PVE}_A$, the proportion of variance in each mediator that can be explained by $A$, among those mediators that are affected by $A$; $\text{PVE}_{IE}$, the proportion of variance of $Y$ that is explained by the total mediation effect; and $\text{PVE}_{DE}$, the proportion of variance of $Y$ that is explained by the direct effect of $A$ on $Y$. Results for varying $\text{PVE}_{IE}$ are presented here while results for varying $\text{PVE}_{DE}$ and $\text{PVE}_A$ are included in the supplement (S3–S6 Figs). In addition to the high-dimensional mediation methods, we include a one-at-a-time method [52] in which the mediators are assessed individually using linear regression. We evaluate the methods by their true positive rate (TPR) for detecting active

mediators, their mean squared error (MSE) for estimating the contributions of active mediators, and their percent relative bias for estimating the global indirect effect.

## Detection of active mediators

Our first evaluation metric is TPR, which is the proportion of the true active mediators the method successfully detected on simulated data. In Fig 3, we show the mean TPR over 100 simulated datasets, with an empirical 95% confidence interval (CI), for both the Group 1 methods and the one-at-a-time approach. To choose signifiance cutoffs for discriminating active mediators from inactive, we used a thresholding procedure within each dataset and each method that fixed the false discovery rate (FDR) below 10% (see Methods). For the non-sparse setting, in which every mediator is active, we show the mean TPR for detecting mediators whose $(\boldsymbol{\alpha_a})_j$ and $(\boldsymbol{\beta_m})_j$ were *both* sampled from Normal(0,1) rather than Normal(0,0.2$^2$). We focus on TPR but not false positive rate (FPR) because the FDR correction was highly conservative, and the mean FPR ranged only from 0 to 5.0x10$^{-4}$ across all settings and methods.

For a sample size of 2,500 and a PVE$_{IE}$ of 0.10, the most powerful method in the baseline setting was BSLMM (mean TPR: 0.45; CI: 0.25–0.63), whose average TPR was 40% higher than that of HDMA, the second-best method. BLSMM also performed best when PVE$_{IE}$ was 0.05 (mean TPR: 0.25; CI: 0.02–0.48), but to a lesser degree, outperforming HDMA by only 13%. BSLMM remained the best method, and HDMA the second best, no matter the signal strength or the degree of correlations, but performed poorly when the signals were non-sparse. In the setting with 1,000 observations, PVE$_{IE}$ set to 0.05, and non-sparse signals, the best-performing method was HIMA (mean TPR: 0.09; CI: 0.05–0.10), its average TPR 3.3 times higher than that of BSLMM, which performed worst.

## Estimation of contributions of active mediators

We now assess the MSE of the methods for estimating mediation contributions of active mediators relative to the one-at-a-time approach. In Fig 4, we show the relative MSE (rMSE) for estimating mediation contributions among the mediators that were either active (in the baseline and high-correlation settings) or had $(\boldsymbol{\alpha_a})_j$ or $(\boldsymbol{\beta_m})_j$ sampled from the larger-variance distribution (in the non-sparse setting). In the baseline setting with 2,500 observations, the best-performing method when the mediation signal was strong was BSLMM, whose mean rMSE of 0.59 (CI: 0.13–1.51) was 24% lower than that of HDMA, the second-best method. However, when the PVE$_{IE}$ was reduced to 0.05 or the sample size reduced to 1,000, the best-performing method was either HDMA or MedFix, with MedFix (mean rMSE: 0.79; CI: 0.31–1.53) performing 61% better than BSLMM after reducing both. Similar trends were observed for the high-correlation and non-sparse settings. Relative MSE for inactive mediators is provided in the supplement (S5 Fig).

## Estimation of global indirect effect

Lastly, Fig 5 shows the percent relative bias for estimating the global mediation effect, $\boldsymbol{\alpha_a}^{\mathbf{T}}\boldsymbol{\beta_m}$. We use the same methods as in Figs 3 and 4 along with the Group 2 methods PCMA and HILMA, which obtain an estimate of the global indirect effect without directly fitting the original mediation model. In the baseline setting with 2,500 samples, the best performer when PVE$_{IE}$ was 0.10 was HILMA, whose mean relative bias of 9% (CI: 0.6% - 20.8%) was 40% lower than that of HDMA, the second-best. When the PVE was reduced to 0.05, the best-performing method was MedFix (mean relative bias: 20.5%; CI: 1.0% - 43.8%), which outperformed HILMA by only 7%. We observed similar results for a sample size of 1,000 and high-

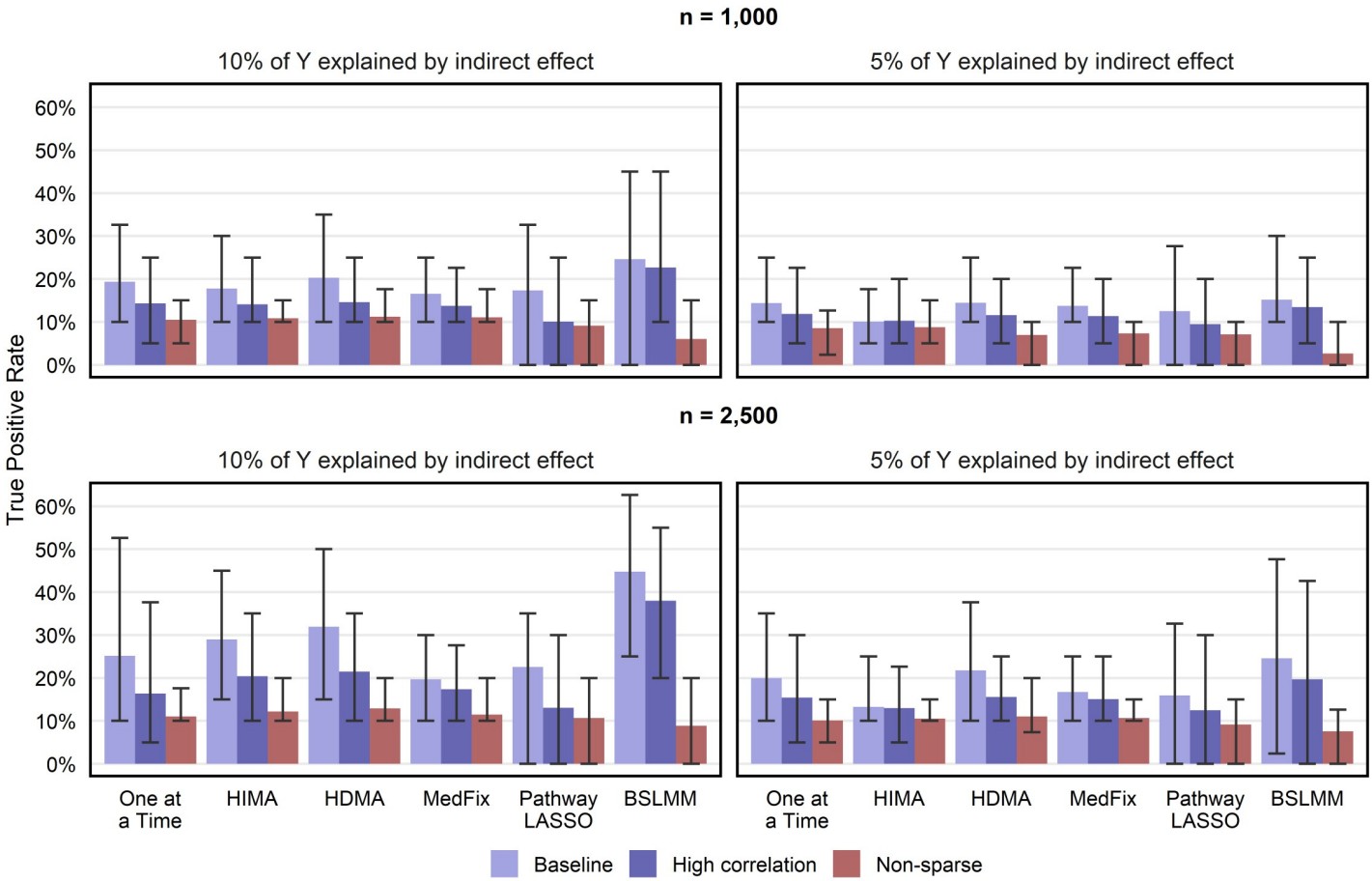

**Fig 3. True positive rate for detecting mediation signals at a false discovery rate of 10%.** Value shown is the mean TPR across 100 simulated data replicates, with intervals representing the inner 95% range. False discovery proportion was capped below 10% by a proper choice of the p-value threshold (one-at-a-time, HIMA, HDMA, MedFix), posterior inclusion probability threshold (BSLMM), or tuning parameter (pathway LASSO).

correlations. In the non-sparse settings, where the biases tended to be much higher, the best performing methods were either PCMA or HDMA.

## DNAm data analysis results from MESA

On an epigenetic dataset with 402,339 CpG sites, we applied SPCMA, HDMM, and every method from our simulation study to infer whether the association between SES and HbA1c is mediated by changes DNAm. For SES, we used a binary variable representing low education level (i.e., education below a 4-year degree), and for DNAm we used M-values [54]. All variables (including methylation values and CpGs) were standardized before analysis. Since the methods are incapable of handling so many CpG sites at once, we reduced our scope to include only the 2,000 sites with the strongest association with low SES based on linear mixed model p-values (see Materials and Methods). Our final dataset contained these 2,000 CpG sites and 963 samples.

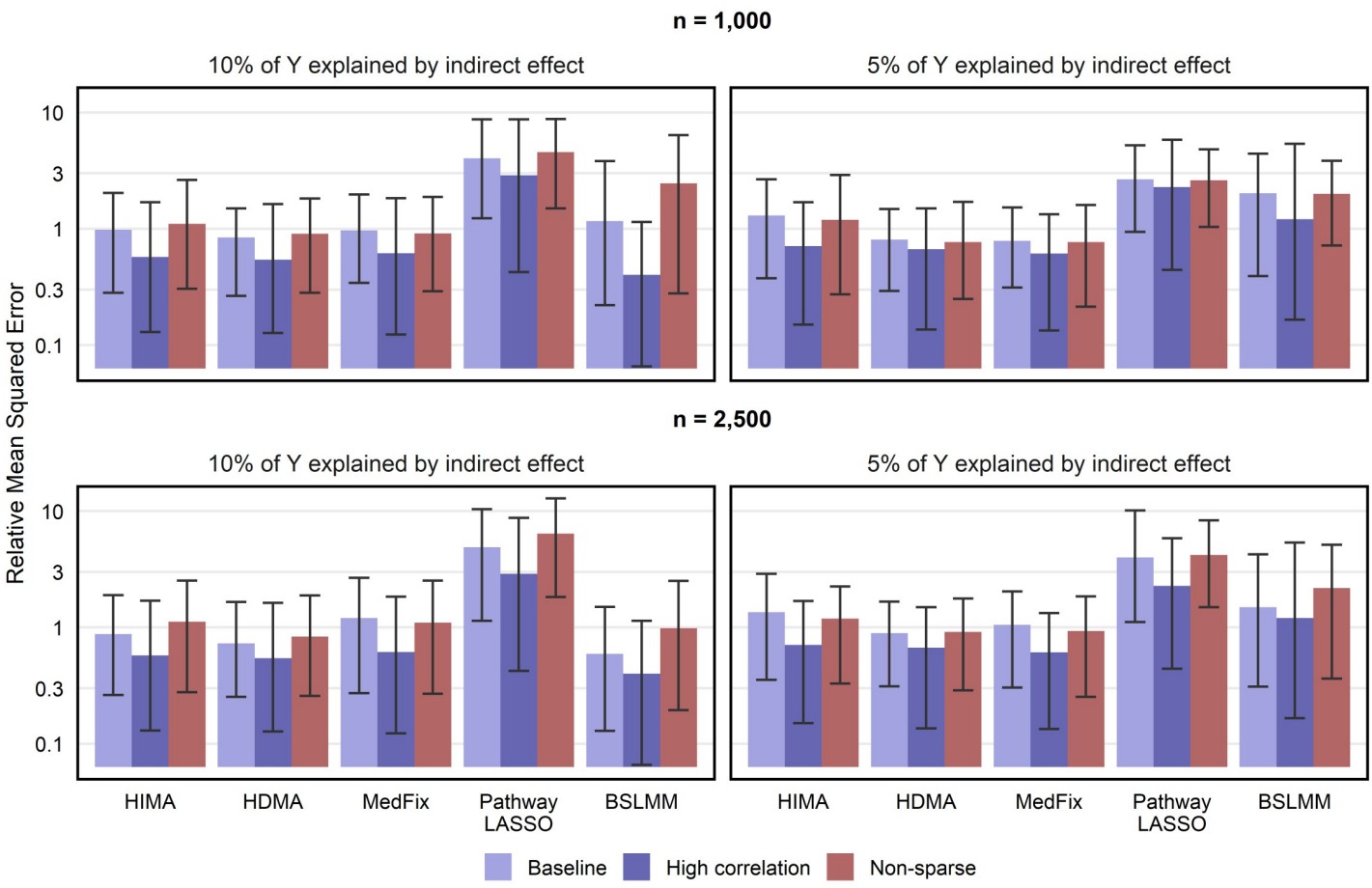

**Fig 4. MSE in estimating mediation contributions of active mediators, relative to one-at-a-time method.** Y-axis is on a $\log_{10}$ scale. Value shown is the mean of the relative mean-squared error for estimating mediation contributions among active mediators (relative to the one-at-a-time approach) across 100 simulated data replicates, with intervals representing the inner 95% range.

### Identification of noteworthy CpG sites

We identified CpG sites that potentially mediated the relationship between low SES and HbA1c using methods from Group 1. In HIMA, HDMA, MedFix, and pathway LASSO, which involve feature selection, we describe a CpG site to be "active" if its estimated mediation contribution is not zero; whereas in BSLMM, we do so if the estimated posterior inclusion probability is not zero (see Materials and Methods). We also included a one-at-a-time method in which the CpG sites were assessed individually with linear mixed models, identifying active mediators with the joint significance test [52]. Out of 2,000 CpG sites, HIMA found 3 sites to be noteworthy, HDMA found 11, MedFix found 3, pathway LASSO found 141, and BSLMM found 3, amounting to 144 unique CpG sites in total. The one-at-a-time method identified zero CpG sites as noteworthy at an FDR threshold of 10%. Eleven CpG sites were identified as noteworthy by at least two of the methods (Table 1). Among these 11, the estimated mediation contributions were similar across methods in direction and size except for BSLMM, for which the estimates were an order of magnitude smaller than the others but in the same direction.

Some of these CpG sites are on or nearby genes that are potentially related HbA1c. Site cg10508317 is in the body of the *SOCS3* gene, for which a rich body of literature has established

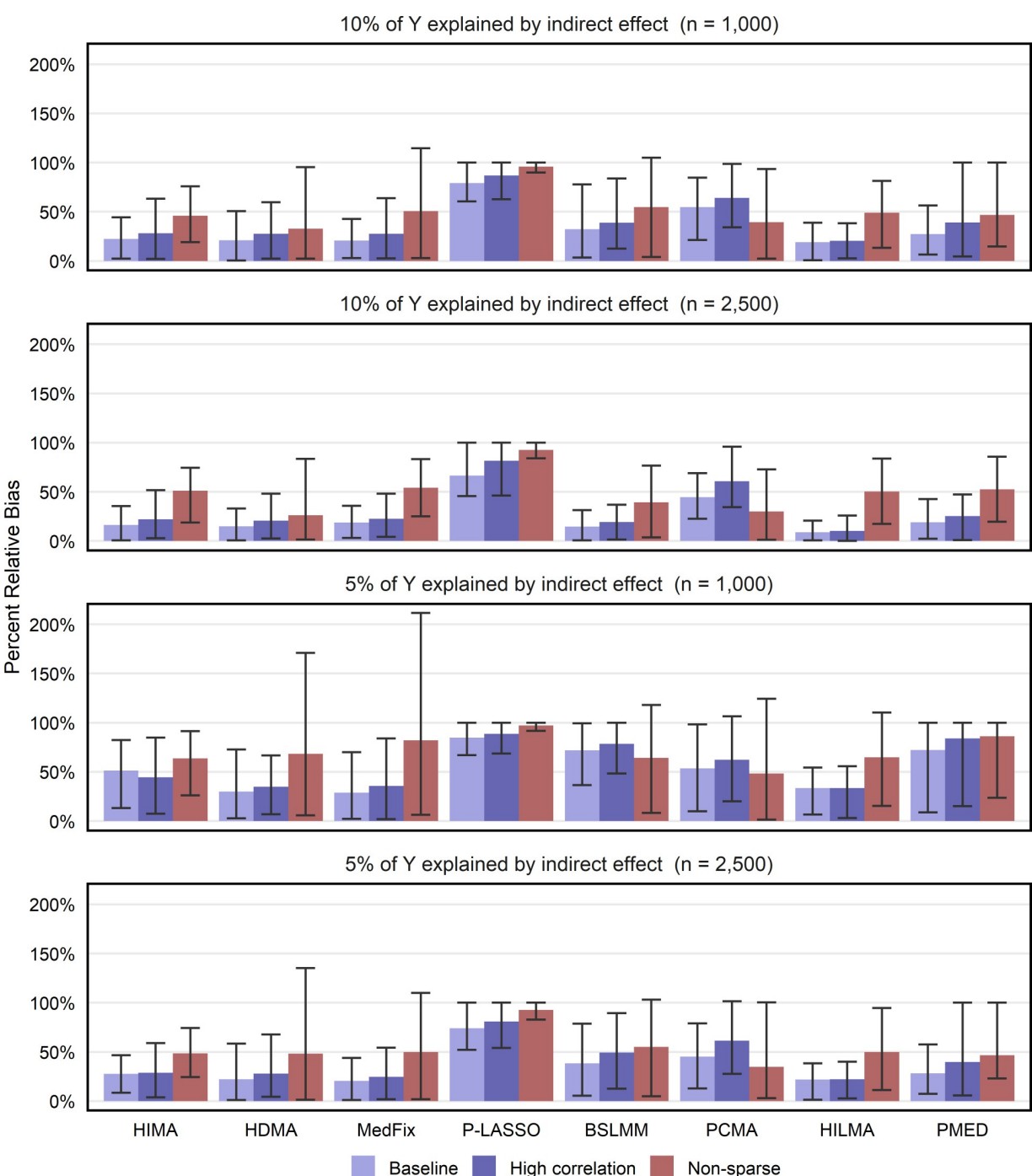

**Fig 5. Percent relative bias in estimated global indirect effect.** Value shown is the mean of the percentage relative bias in estimating the global mediation effect across 100 simulated data replicates, with intervals representing the inner 95% range.

links between overexpression and insulin resistance [56], and has previously been been identified in MESA as a mediator between adult SES and BMI [11] and adult SES and HbA1c [18] based on single-mediator analysis. Site cg01288337, which is in the body of the *RIN3* gene, has been identified in MESA as a potential mediator between adult SES and HbA1c based on one-at-a-time analysis as well [18]. The *RIN3* gene itself is proximal to the *SLC24A4* gene, both of

**Table 1. Estimated contributions of noteworthy CpG sites on the mediation pathway between low education and HbA1c.**

| CpG Name | Chromosome | Nearby Gene (s) | USCS RefGene Group | Univariate (0 sites identified) | HIMA (3 sites identified) | HDMA (11 sites identified) | MedFix (3 sites identified) | Pathway LASSO (141 sites identified) | BSLMM (3 sites identified) |
|---|---|---|---|---|---|---|---|---|---|
| cg10508317 | 17 | SOCS3 | Body | $3.48 \times 10^{-2}$ | $1.59 \times 10^{-2}$* | $3.56 \times 10^{-2}$* | $2.90 \times 10^{-2}$* | $2.35 \times 10^{-2}$* | $0.25 \times 10^{-2}$ |
| cg01288337 | 14 | RIN3 | Body | $3.35 \times 10^{-2}$ | $1.47 \times 10^{-2}$* | $2.82 \times 10^{-2}$* | $2.70 \times 10^{-2}$* | $4.43 \times 10^{-2}$* | $0.21 \times 10^{-2}$ |
| cg10244976 | 16 | LMF1 | Body | $3.00 \times 10^{-2}$ | 0 | $2.78 \times 10^{-2}$* | 0 | $2.23 \times 10^{-2}$* | $0.19 \times 10^{-2}$ |
| cg07516252 | 14 | REC8 | TSS200 | $2.72 \times 10^{-2}$ | 0 | $2.24 \times 10^{-2}$* | 0 | $2.26 \times 10^{-2}$* | $0.26 \times 10^{-2}$ |
| cg07571519 | 10 | C10orf105; CDH23 | 3'UTR; Body | $2.53 \times 10^{-2}$ | $0.33 \times 10^{-2}$* | $3.67 \times 10^{-2}$* | $1.47 \times 10^{-2}$* | $2.81 \times 10^{-2}$* | $0.21 \times 10^{-2}$ |
| cg23079012 | 2 | LINC00299 | Body | $2.27 \times 10^{-2}$ | 0 | $1.99 \times 10^{-2}$* | 0 | $1.98 \times 10^{-2}$* | $0.29 \times 10^{-2}$ |
| cg01587454 | 8 | DCAF4L2 | 1stExon | $1.77 \times 10^{-2}$ | 0 | $2.10 \times 10^{-2}$* | 0 | $1.99 \times 10^{-2}$* | $0.38 \times 10^{-2}$ |
| cg27527503 | 4 | HADH | TSS1500 | $1.75 \times 10^{-2}$ | 0 | $1.86 \times 10^{-2}$* | 0 | $1.27 \times 10^{-2}$* | $0.23 \times 10^{-2}$ |
| cg25891647 | 11 | GRAMD1B | Body | $-1.27 \times 10^{-2}$ | 0 | $-3.42 \times 10^{-2}$* | 0 | $-3.02 \times 10^{-2}$* | $-0.33 \times 10^{-2}$ |
| cg08473752 | 17 | NLK | Body | $-0.70 \times 10^{-2}$ | 0 | $-2.34 \times 10^{-2}$* | 0 | $-2.32 \times 10^{-2}$* | $-0.22 \times 10^{-2}$ |
| cg12644059 | 15 | BLM | N/A[1] | $-0.03 \times 10^{-2}$ | 0 | $-2.31 \times 10^{-2}$* | 0 | $-1.84 \times 10^{-2}$* | $-0.22 \times 10^{-2}$ |

*Selected as noteworthy by given method

[1]CpG site cg12644059 is 3.240kb from the final base pair of the BLM gene

Includes all CpG sites that were selected as having a noteworthy mediation contribution by at least two of the implemented methods out of 2,000 CpG sites in total. All estimates are adjusted for age, sex, race, and the estimated proportions of residual non-monocytes as fixed effects, along with methylation chip and position as random effects to address potential batch effects. Note that for HIMA, HDMA, MedFix, and pathway LASSO, which fit high-dimensional regression models, we used additional pre-screening to reduce the number of mediators in advance to only $n/\log(n) \approx 141$ CpG sites, which is the approach recommended by the HIMA and HDMA authors and helps with statistical and computational efficiency (see Materials and Methods). Pathway LASSO selected all of these.

which have been linked to brain glucose metabolism in human population studies [57]. In addition, site cg27527503 is in the promoter region of the *HADH* gene, which is differentially expressed with respect to diabetes status [58] and is a primary driver of hyperinsulinism [59] and hyperinsulinaemic hypoglycemia [60]. A Venn diagram of genes identified by the methods is included in the supplement (S5 Fig), and results for every noteworthy CpG site are provided in the supplement (S1 File).

## Global mediation through DNAm

Next, we estimated the direct effect of low education on HbA1c, the global indirect effect of low education on HbA1c through DNAm, and the total effect of low education on HbA1c using the Group 1 methods HIMA, HDMA, MedFix, pathway LASSO, and BSLMM, as well as the Group 2 methods PCMA, SPCMA, and HILMA (Table 2). Results across methods varied considerably, with the estimated global indirect effect ranging from 0 in PMED to 0.17 in SPCMA. The estimated total effect ranged from 0.03 (HILMA) to 0.198 (HIMA, HDMA, and MedFix). Despite the variability in the estimated global indirect effect, some of the other methods were consistent, with HDMA, BSLMM, pathway LASSO, PCMA, and SPCMA all estimating the global indirect effect to be close to 0.15. The variability in the estimated indirect effect and estimated total effect led to variability in the proportion mediated as well, from 17.1% in HIMA to 100% in HILMA.

## Additional findings

In addition to estimating the global indirect effect, the method SPCMA is also able to identify potentially-mediating CpG sites in groups. It does so by linearly combining the mediators using sparse principal component-defined weights, then evaluating the resulting principal

**Table 2. Estimated effects in the mediation mechanism from low education to DNAm to HbA1c.**

| Method | Estimated Global Indirect Effect | Estimated Direct Effect | Estimated Total Effect | Estimated Proportion Mediated |
|---|---|---|---|---|
| HIMA | 0.03 | 0.16 | 0.20 | 0.17 |
| HDMA | 0.13 | 0.07 | 0.20 | 0.65 |
| MedFix | 0.07 | 0.13 | 0.20 | 0.36 |
| BSLMM | 0.14 | 0.05 | 0.18 | 1.00 |
| Pathway LASSO | 0.13 | 0.05 | 0.18 | 0.74 |
| PCMA | 0.15 | 0.02 | 0.17 | 0.91 |
| SPCMA | 0.17 | 0.00 | 0.17 | 1.00 |
| HILMA | 0.03 | 0.00 | 0.03 | 1.00 |
| PMED | 0.00 | 0.20 | 0.20 | 0.00 |

All estimates are adjusted for age, sex, race, and the estimated proportions of residual non-monocytes as fixed effects, along with methylation chip and position as random effects to address potential batch effects. We provide only point estimates, not interval estimates, because some of the methods are either not capable of producing interval estimates or do not provide the code for producing them in their software. Note also that for HIMA, HDMA, MedFix, and pathway LASSO, we used additional screening to reduce the number of mediators in advance for the sake of statistical and computational efficiency, so only $n/\log(n) \approx 141$ CpG sites were seen by the multivariable outcome model rather than 2,000.

components as mediators themselves [42]. However, out of 100 computed principal components, only three of them had significant mediation contributions after 10% FDR correction, the first representing a linear combination of 762 CpG sites, the second a combination of 782 sites, and the third a combination of 797 sites. Since the transformed mediators are functions of so many CpG sites at once, one cannot make claims about which particular CpG sites are active mediators, but the method still provides insight to whether there is statistical mediation at all.

We conclude our analysis by applying HDMM, a method from Group 3. Unlike the methods in Groups 1 and 2, HDMM cannot be used to estimate the global indirect effect from the proposed mediation structure, nor to estimate the mediation contributions of specific CpG sites. Rather, HDMM uses a likelihood-based approach to compute "directions of mediation", which are weights that can be used to linearly combine the observed mediators into unobserved, latent mediators that replace the observed mediators in the mediation models. The estimated effect of the first latent mediator on average HbA1c was 0.13, the estimated total effect 0.71, and the proportion mediated 0.715. The three CpG sites with the largest directions of mediation were cg01288337 (0.36) on the *RIN3* gene, cg16162970 (-0.22) near the *PACS2* gene, and cg25891647 (-0.21) on the *GRAMD1B* gene; the first and last of which were among the 11 CpG sites identified by other methods in Table 1. Although the size and direction of these estimates are not interpretable, they offer evidence that these CpG sites are potentially involved in mediation.

## Discussion

In this study, we reviewed and evaluated eight statistical methods for performing mediation analysis with high-dimensional DNAm data, so that researchers in epigenetics have the information they need to choose the most appropriate method for their data sample, subject matter, and research objectives. In extensive simulations, we found that the most powerful method for identifying active mediators was generally BSLMM, with HDMA as a close comparator. However, BSLMM performed poorly in settings where the mediation signals were non-sparse. No method was uniformly better than the others at estimating the mediation contributions though Pathway LASSO appeared to be a sub-optimal choice. For estimating the global indirect effect, the best-performing method was HILMA in sparse mediation settings and PCMA or HDMA

in non-sparse mediation settings. In simulation settings where the effects are strictly non-negative, BSLMM tended to perform best for detecting active mediators and estimating their mediation contributions, while HILMA was again the strongest method for estimating the global mediation effect (S1 Fig). In simulation scenarios with an unmeasured confounder, the performance of the multiple-mediator methods became worse as the severity of the confounding effects increased, in terms of estimating the global mediation effect or inferring the mediation contributions (S2 Fig). However, the relative performance of these methods compared to the one-at-a-time approach improved substantially with more confounding, which emphasizes the importance of evaluating the mediators simultaneously rather than one-by-one. Our comparison of the scalability of the methods revealed that HIMA, HDMA, MedFix, PMED, and PCMA were easily scalable to large datasets (e.g., $n = 1,000$ and $p = 2,000$), whereas SPCMA and pathway LASSO were computationally expensive (Section 5 in S1 Text).

On DNAm data from MESA, 11 CpG sites were selected by at least two of the methods as mediators between low SES and HbA1c level. Of the many genes related to these sites, *SOCS3*, *RIN3*, and *HADH* have the strongest potential biological connections to HbA1c [56–58,60–62], which contributes to the already rich literature on DNAm as a mediator between the exposome and health outcomes. Moreover, the methods generally produced similar estimates of the mediation contributions, with the exception of BSLMM. It is possible that since estimated from BSLMM is non-sparse, the estimated mediation contributions end up severely shrunken compared to the methods that directly select features.

Estimates of the global indirect effect were highly variable. Part of this can be explained by the fact that HDMA, MedFix, HIMA, and pathway LASSO are sparse models that can set mediation contributions to be exactly zero, resulting in a rigid and unstable estimation of the global indirect effect. The method HILMA, which is built specifically for estimating the global indirect effect and direct effect, produced estimates that were sharply different than the other methods, possibly because our simulations indicated that it struggled in non-sparse mediation settings.

In practice, the optimal method for mediation analysis with high-dimensional mediators will depend both on the data and the objective. If the goal is to identify specific CpG sites that are involved in mediation, one preferred method may be HDMA, which performed well at detecting active mediators in our simulations and was not overly conservative when applied to the observed DNAm data. If one's focus is the global indirect effect, our simulations suggested that the optimal method is HILMA; but considering the variability we observed in our DNAm analysis, it may be worthwhile to apply BSLMM and HDMA as well to ensure the results are robust. If the results of multiple methods disagree substantially, it may be difficult to say with confidence which is closest to the truth, and the estimates should be interpreted with caution. Next, if there is interest in latent, unmeasured mediators, either HDMM or LVMA is worth attempting, although HDMM is simpler computationally. A detailed decision tree to aid the user for selecting the optimal method is included in Fig 6.

Some strengths of our study include its broad coverage of the available methods, the breadth of its simulation settings, and the comprehensive set of evaluation criteria. Our analysis of real DNAm data is especially essential because it elucidates the potential limitations of using these methods in practice, as it is impossible to incorporate the full complexity of real data sources into contrived simulation settings. Another strength of our study is the presentation of an R package, as the lack of readily available, centralized software for implementing methods for high-dimensional mediation analysis is a potential reason for their so far limited permeation into epigenetic research. We are hopeful that our package, *hdmed*, will facilitate and encourage the application and adoption of these methods to epigenetic datasets in future studies.

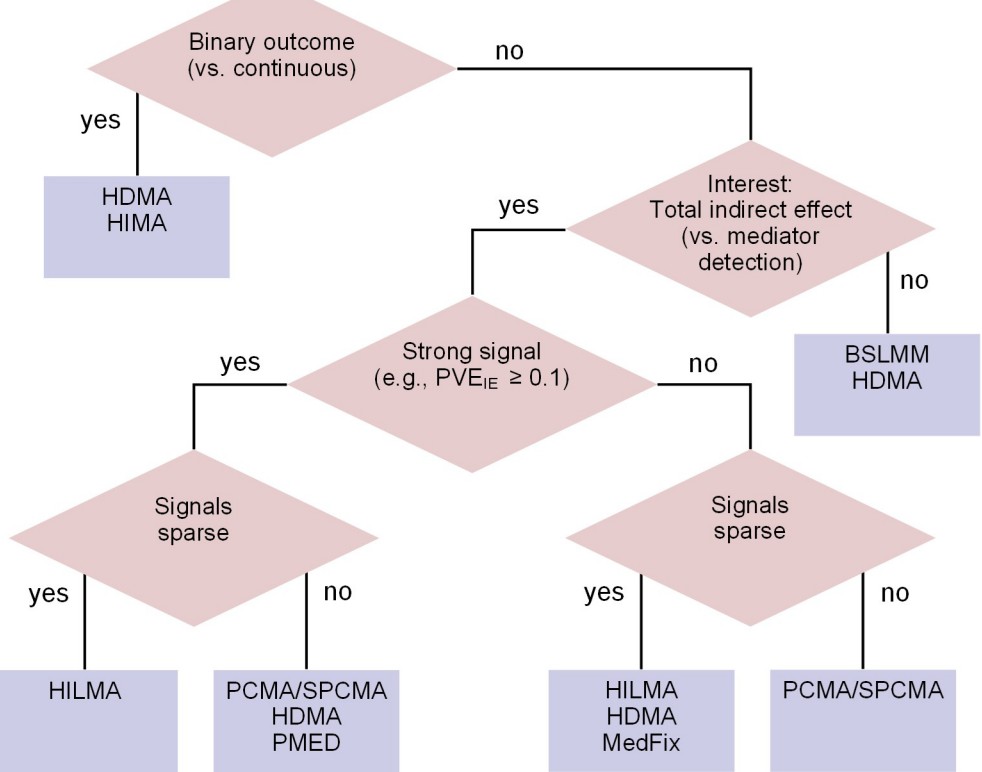

**Fig 6. Decision tree for selecting a high-dimensional mediation analysis.**

However, our study also has weaknesses. First, since DNAm measurements and HbA1c data were collected concurrently, and represent only single time points, we cannot interpret the parameters we have estimated as causal effects. Although it would be optimal to address our research question longitudinally—with measurements at multiple time points—there is a dearth of mediation analysis methods which can handle that type of data, and longitudinal mediation analysis with high-dimensional mediators should be a focus of future methodological development. Second, the validity of the mediation analysis depends on the strong assumption that the causal mechanism is correctly specified—that is, that the exposure affects the mediators, that the mediators affect the outcome, and that confounding of this relationship is accounted for by the model. If there is unmeasured confounding of the causal pathway, or if some of the variables treated as mediators are, in fact colliders, the parameters of the high-dimensional mediation model become difficult to interpret, and the estimate of the global indirect effect may be highly biased. Though recent work by [63] has directly considered the issue of unmeasured confounding in a high-dimensional mediation model, the issue of collider bias is an important area for future research.

Third, we limited our analysis to the situation that $Y$ and $M$ are continuous, that $M$ and $A$ do not interact, and that only one $A$ is of interest. However, we note that the methods HIMA and HDMA can also be applied to identify active mediators when $Y$ is binary, while PCMA can be applied to infer the global indirect effect when there is $A$-$M$ interaction in the outcome model. MedFix, along with the simultaneously-proposed MedMix (mediation analysis with mixed effect model by Zhang (2021)) can be applied when both the exposures and mediators are high-dimensional, while Huang and Vanderweele (2014) proposed a variance component

test of the global indirect effect when only *A* is high-dimensional [64]. If one has prior knowledge that the signs of the outcome model coefficients are in the same direction, a reasonable approach might be to use sign-constrained optmization rather than standard penalties such as the LASSO [65]. In terms of data type, methods that can accommodate non-continuous *Y* and *M* are in general scarce, and represent an important direction for future methodological development. As the landscape of methods for high-dimensional mediation analysis continues to expand, future review studies should consider exploring additional mediation settings (in presence of non-linearity, interaction) for which statistical methods are continuing to become available.

## Supporting information

**S1 Text. Methods for Mediation Analysis with High-Dimensional DNA Methylation Data: Possible Choices and Comparison.**
(PDF)

**S1 File. Estimated mediation contributions of CpG sites identified as mediators by any method.**
(XLSX)

**S2 File. Numerical results underlying the figures from the simulation study.**
(XLSX)

**S3 File. Zip file of computer code for performing the simulation study.**
(ZIP)

**S1 Fig. Results for simulations with strictly non-negative effects.** (A) True positive rate for detecting active mediators. (B) Relative mean squared error for estimating the mediation contributions of active mediators, relative to the one-at-a-time method. (C) Relative mean squared error for estimating the mediation contributions of inactive mediators, relative to the one-at-a-time method. (D) Percent relative bias for inferring the global mediation effect. The simulation settings for were created by taking the absolute values of the exposure-mediator and mediator-outcome effects in the original baseline simulation settings, which had four different proportion-of-variance-explained (PVE) settings: (1) $PVE_A = 0.2$, $PVE_{DE} = 0.1$, $PVE_{IE} = 0.1$; (2) $PVE_A = 0.1$, $PVE_{DE} = 0.1$, $PVE_{IE} = 0.1$; (2) $PVE_A = 0.2$, $PVE_{DE} = 0.05$, $PVE_{IE} = 0.1$. (4) $PVE_A = 0.2$, $PVE_{DE} = 0.1$, $PVE_{IE} = 0.05$.
(PNG)

**S2 Fig. Results for simulations with an unmeasured confounder *U*.** (A) True positive rate for detecting active mediators. (B) Mean squared error for inferring the mediation contributions of *active* mediators. (C) Relative mean squared error for inferring the mediation contributions of *active* mediators, relative to the one-at-a-time method. (D) Percent relative bias for inferring the global mediation effect.
(PNG)

**S3 Fig. True positive rate for detecting mediation signals at a false discover rate of 10%.** Mean true positive (TPR) rate and 95% empirical confidence interval for detecting active mediators in 100 simulated datasets. In the baseline and high-correlation-among-mediators settings, TPR is for distinguishing mediators which contribute to the global mediation effect from those which do not, whereas in the non-sparse setting, where all mediators contribute, TPR is for distinguishing mediators whose contributions were sampled from a high-variance distribution from those whose contributions were sampled from a low-variance distribution.

False discovery rate was capped below 10% by a proper choice of the p-value threshold (one-at-a-time, HIMA, HDMA, MedFix), posterior inclusion probability threshold (BSLMM), or method tuning parameter (P-LASSO). PVE(A): Percent of variance in $Y$ explained by the exposure. PVE(IE): Percent of variance in $Y$ explained by the indirect effect. PVE(DE): Percent of variance in $Y$ explained by the direct effect.
(PNG)

**S4 Fig. MSE in estimating mediation contributions of active mediators, relative to one-at-a-time method.** Mean relative mean squared error (rMSE) and 95% empirical confidence interval for estimating mediation contributions among active mediators in 100 simulated datasets, relative to the one mediator at a time method. Y-axis is on a $\log_{10}$ scale. For the baseline and high-correlation-between-mediators settings, active mediators are those which contribute to the global mediation effect, whereas in the non-sparse setting, where all mediators have some contribution, active mediators are those whose contributions were sampled from a distribution with large variance instead of small. PVE(A): Percent of variance in $Y$ explained by the exposure. PVE(IE): Percent of variance in $Y$ explained by the indirect effect. PVE(DE): Percent of variance in $Y$ explained by the direct effect.
(PNG)

**S5 Fig. MSE in estimating mediation contributions of inactive mediators, relative to one-at-a-time method.** Mean relative mean squared error (rMSE) and 95% empirical confidence interval for estimating mediation contributions among inactive mediators in 100 simulated datasets, relative to the one mediator at a time method. Y-axis is on a $\log_{10}$ scale. For the baseline and high-correlation-between-mediators settings, inactive mediators are those which have no mediation contribution, whereas in the non-sparse setting, where all mediators have some contribution, inactive mediators are those whose contributions were sampled from a distribution with small variance instead of large. The method pathway LASSO is excluded from this figure because for multiple settings it had rMSEs of exactly zero. This happened because pathway LASSO tended to be highly conservative and successfully assigned inactive mediators to have no effect. PVE(A): Percent of variance in $Y$ explained by the exposure. PVE(IE): Percent of variance in $Y$ explained by the indirect effect. PVE(DE): Percent of variance in $Y$ explained by the direct effect.
(PNG)

**S6 Fig. Percent relative bias in estimated global indirect effect.** Mean percentage relative bias in estimating the global mediation effect across 100 simulated data replicates, with intervals representing the inner 95% range. PVE(A): Percent of variance in $Y$ explained by the exposure. PVE(IE): Percent of variance in $Y$ explained by the indirect effect. PVE(DE): Percent of variance in $Y$ explained by the direct effect.
(PNG)

**S7 Fig. Genes containing or near CpG sites selected as active mediators between low education and HbA1c by methods for high-dimensional mediation analysis.** CpG sites were linked to genes using R Bioconductor package "IlluminaHumanMethylation450kanno. ilmn12.hg19". Additional genes detected by Pathway LASSO listed in supplementary S1 File.
(JPG)

**S1 Table. Summary of methods for high-dimensional mediation analysis.**
(PDF)

**S2 Table. Complete list of primary simulation settings.**
(PDF)

**S3 Table. Complete list of additional simulation settings.**
(PDF)

## Acknowledgments

The authors wish to thank the MESA staff and participants.

## Author Contributions

**Conceptualization:** Dylan Clark-Boucher, Xiang Zhou, Jennifer A. Smith, Bhramar Mukherjee.

**Data curation:** Yongmei Liu, Belinda L. Needham.

**Formal analysis:** Dylan Clark-Boucher.

**Funding acquisition:** Jennifer A. Smith, Bhramar Mukherjee.

**Methodology:** Dylan Clark-Boucher, Xiang Zhou, Jiacong Du, Jennifer A. Smith, Bhramar Mukherjee.

**Supervision:** Jennifer A. Smith, Bhramar Mukherjee.

**Visualization:** Dylan Clark-Boucher, Bhramar Mukherjee.

**Writing – original draft:** Dylan Clark-Boucher, Xiang Zhou, Jennifer A. Smith, Bhramar Mukherjee.

**Writing – review & editing:** Dylan Clark-Boucher, Xiang Zhou, Jiacong Du, Yongmei Liu, Belinda L. Needham, Jennifer A. Smith, Bhramar Mukherjee.

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
