## [Decision Letter · Decision Letter 0]

21 Jun 2023

Dear Dr Mukherjee,

Thank you very much for submitting your Research Article entitled 'Methods for mediation analysis with high-dimensional DNA methylation data: Possible choices and comparisons' to PLOS Genetics.

The manuscript was fully evaluated at the editorial level and by independent peer reviewers. The reviewers appreciated the attention to an important problem, but raised some substantial concerns about the current manuscript. Based on the reviews, we will not be able to accept this version of the manuscript, but we would be willing to review a much-revised version. We cannot, of course, promise publication at that time.

If you decide to revise the manuscript for further consideration at PLOS Genetics, please aim to resubmit within the next 60 days, unless it will take extra time to address the concerns of the reviewers, in which case we would appreciate an expected resubmission date by email to plosgenetics@plos.org.

We are sorry that we cannot be more positive about your manuscript at this stage. Please do not hesitate to contact us if you have any concerns or questions.

Yours sincerely,

Zoltán Kutalik, PhD

Academic Editor

PLOS Genetics

John Greally

Section Editor

PLOS Genetics

As the authors can see the reviewers raised some important points, but appreciate the usefulness of the findings and the addressed topic. Beyond their comments, I'd like to ask the authors to address the point of unmeasured confounders: in the simulations it is always assumed that there is no unmeasured confounder and the linear model fit reflects causation. Also, what happens if variables treated as mediators are actually colliders? Such simulation scenarios (including model misspecifications) and the presence of unmeasured confounding factors should be explored and commented on in the Discussion.

Reviewer's Responses to Questions

**Comments to the Authors:**

Reviewer #1: The manuscript is a timely evaluation and comparison of the methods for high dimensional mediation with a single exposure, continuous mediators and continuous outcome. The R package is also a valuable resource for the practitioners working with high dimensional mediation analysis. While the paper focuses on methylation data, it is expected to benefit the researchers in other areas as well. The following are my specific comments.

1.Correlation: the issue of “correlations” or correlated mediators is mentioned in many places in this manuscript. The mediators may be correlated for different reasons. They can be causally related, or their noise terms may be correlated, or they are conditionally independent given the exposure and the confounders, and merely marginally correlated due to the exposures and the confounders. The authors did not make any distinction between them, which may unintentionally mislead the readers. One such example (along many) is line 94-96 on page 5. It reads “Instead, so that we leverage these correlations rather than ignore them, the preferred approach is to evaluate the mediators jointly through a single, multivariable statistical model.” This is largely not true. Using the multivariate outcome model only addresses the marginal correlation due to their dependence on common exposure and confounders. Most of the surveyed methods assume that the mediators are not causally interdependent, and some of the inference procedures further assume independent noise for mediators. The authors need to clarify what types of correlations that these methods can “leverage”, and what their common limitations are.

2.Interpretations of the mediation contribution: This is partially related to the correlation issue. The authors have presented the causal assumptions for the total mediation effect but not the mediation effect of the individual mediators. Instead, they say that they cannot be interpreted as a causal effect through the jth mediator. This is true in general. But it will be more helpful if the authors can present the assumptions under which alpha_j*beta_j can be interpreted as a causal effect through the jth mediator. I believe that some of the surveyed papers have presented such assumptions. The authors also need to clarify further the meaning of “mediation contribution” and its limitations. For example, if a mediator’s mediation effect is completed mediated by other mediators, it may not be significant in the outcome model, but it is still “active” somewhere in the causal network among the mediators.

3.The simulation setting appears to resemble a setting with inconsistent mediation in which half of the mediators have positive mediation effects and the other half have negative mediation effects, and they partially cancel each other out. While it is common in multivariate mediation analysis, it will be better to include a case where the mediation effects of the individual mediators are more consistent. It means that the direct effect, total effect, and the mediation effects of most true mediators have the same sign.

4.It will benefit the readers the most, if these methods are evaluated using a real dataset that they have direct access to, instead of behind dbGaP wall.

5.What are the average or median magnitudes of the correlations in the moderate and high correlation simulation settings? They should have meaningful differences.

6.The following paper is a method similar to HILMA published in JASA in 2022. It is faster than HILMA, and the application involves methylation data. It also reports a set of “important mediators” based on the variable selection of the outcome model. But it is not exactly mediator selection and there is no inference for it. They provided code and script to reproduce their results. It is NOT my paper.

Guo, X., Li, R., Liu, J., & Zeng, M. (2022). High-dimensional mediation analysis for selecting DNA methylation Loci mediating childhood trauma and cortisol stress reactivity. Journal of the American Statistical Association, 117(539), 1110-1121.

7.The website for the real data https://www.mesanhlbi.org/ may contain typos. The data availability statement should mention that dbGaP application is needed.

8.Line 130 on page 6, can p,q be larger than n?

Reviewer #2: Referee report on “Methods for mediation analysis with high-dimensional DNA methylation data: Possible choices and comparisons”

General comment: In this paper, the authors reviewed and evaluated seven statistical methods of mediation analysis using simulations and high-dimensional DNA methylation (DNAm) data. The authors also centralize the computer codes of all the methods into a single, stand-alone R package. In addition, the authors provide some guidelines for the usage of these statistical methods. In particular, the authors created a decision tree for selecting a high-dimensional mediation analysis.

The following are some detailed comments.

Comments:

1. In introduction, the authors said that “Though several methods for fitting such a model have been presented in the literature, none of them are widely used in analyzing DNAm data.” However, there are some high-dimensional mediation methods that have been used in DNAm data, e.g., HIMA2 in [1].

2. Moreover, HIMA2 is an extension of the HIMA method. The authors have evaluated the HIMA methods. I was wondering whether HIMA2 will perform better if we add it into comparison.

3. In data application, the authors mentioned that they used model (6) to select 2000 CpG sites. But I did not find the model (6) in the paper. Can the authors write or locate the model (6) more clearly?

4. In simulation analysis on Page 16, the authors mentioned not only the MSE in mediation contributions of active mediators but also MSE in mediation contributions of mediators. However, in simulation results, only the MSE in contributions of active

mediators is presented. I was wondering why the MSE in mediation contributions of all mediators is ignored.

5. In the DNAm data analysis, the authors calculated the estimated mediation contributions of each method. What is a clear definition of the estimated mediation contributions?

6. In simulations, the authors generated the exposure, mediators, and outcome from continuous distribution. However, the decision tree for selecting a high-dimensional mediation analysis in Figure 6 says that we should choose HDMA and HIMA methods when the outcome is binary. Should we provide some simulations for the performance of the two methods when the outcome is binary?

References

[1] Perera, C., Zhang, H., Zheng, Y., Hou, L., Qu, A., Zheng, C., Xie, K., and Liu, L. (2022). Hima2: high-dimensional mediation analysis and its application in epigenome-wide dna methylation data. BMC bioinformatics, 23(1):1–14.

**Have all data underlying the figures and results presented in the manuscript been provided?**

Reviewer #1: **No: **Not sure whether it is necessary for this data analysis paper, but I did not see any "numerical data that underlies graphs" in the supplementary

Reviewer #2: None

PLOS authors have the option to publish the peer review history of their article (what does this mean?). If published, this will include your full peer review and any attached files.

Reviewer #1: No

Reviewer #2: No

---

## [Decision Letter · Decision Letter 1]

18 Oct 2023

Dear Dr Mukherjee,

We are pleased to inform you that your manuscript entitled "Methods for mediation analysis with high-dimensional DNA methylation data: Possible choices and comparisons" has been editorially accepted for publication in PLOS Genetics. Congratulations!

Please note the comment from reviewer 2:

"In the responses, the authors said that they included a binary case in Figure 6. But I did not find that case in Figure 6. Is this table explained in the manuscript?" 

Please address this in your editing process, in case there is something missing that would compromise your publication.

Yours sincerely,

Zoltán Kutalik, PhD

Academic Editor

PLOS Genetics

John Greally

Section Editor

PLOS Genetics

Comments from the reviewers (if applicable):

Reviewer's Responses to Questions

**Comments to the Authors:**

Reviewer #1: The revised manuscript has addressed my original concerns.

Reviewer #2: The authors have appropriately addressed most of my comments. I just have the following minor comments.

In the responses, the authors said that they included a binary case in Figure 6. But I did not find that case in Figure 6. Is this table explained in the manuscript?

**Have all data underlying the figures and results presented in the manuscript been provided?**

Reviewer #1: None

Reviewer #2: None

PLOS authors have the option to publish the peer review history of their article (what does this mean?). If published, this will include your full peer review and any attached files.

Reviewer #1: No

Reviewer #2: No

**Data Deposition**

http://datadryad.org/submit?journalID=pgenetics&manu=PGENETICS-D-23-00581R1

**Press Queries**

---

## [Editor Report · Acceptance letter]

30 Oct 2023

PGENETICS-D-23-00581R1 

Methods for mediation analysis with high-dimensional DNA methylation data: Possible choices and comparisons 

Dear Dr Mukherjee, 

We are pleased to inform you that your manuscript entitled "Methods for mediation analysis with high-dimensional DNA methylation data: Possible choices and comparisons" has been formally accepted for publication in PLOS Genetics! Your manuscript is now with our production department and you will be notified of the publication date in due course.

With kind regards,

Judit Kozma

PLOS Genetics

On behalf of:
